# The PMIP4 Last Glacial Maximum experiments: preliminary results and comparison with the PMIP3 simulations

Masa Kageyama[1], Sandy P. Harrison[2], Marie-L. Kapsch[3], Marcus Lofverstrom[4], Juan M. Lora[5], Uwe Mikolajewicz[3], Sam Sherriff-Tadano[6], Tristan Vadsaria[6], Ayako Abe-Ouchi[6], Nathaelle Bouttes[1], Deepak Chandan[7], Lauren J. Gregoire[8], Ruza F. Ivanovic[8], Kenji Izumi[9], Allegra N. LeGrande[10], Fanny Lhardy[1], Gerrit Lohmann[11], Polina A. Morozova[12], Rumi Ohgaito[13], André Paul[14], W. Richard Peltier[7], Christopher J. Poulsen[15], Aurélien Quiquet[1], Didier M. Roche[1,16], Xiaoxu Shi[10], Jessica E. Tierney[4], Paul J. Valdes[9], Evgeny Volodin[17], Jiang Zhu[18]

[1]Laboratoire des Sciences du Climat et de l'Environnement/Institut Pierre-Simon Laplace, UMR CEA-CNRS-UVSQ, Université Paris-Saclay, 91191 Gif-sur-Yvette, France
[2]School of Archaeology, Geography and Environmental Science (SAGES), University of Reading, UK
[3]Max Planck Institute for Meteorology, 20146 Hamburg, Germany
[4]University of Arizona, Tucson, AZ 85721, USA
[5]Yale University, New Haven, CT 06520, USA
[6]Atmospheric and Ocean Research Institute, The University of Tokyo, Kashiwa, Japan
[7]Department of Physics, University of Toronto, 60 St. George Street, Toronto, Ontario M5S1A7, Canada
[8]School of Earth and Environment, University of Leeds, Woodhouse Lane, Leeds, LS2 9JT, UK
[9]School of Geographical Sciences, University of Bristol, University Road, Bristol, BS8 1SS, UK
[10]NASA Goddard Institute for Space Studies, 2880 Broadway, New York, NY 10025, USA
[11]Alfred Wegener Institute, Bremerhaven, Germany
[12]Institute of Geography, Russian Academy of Science, Moscow, Russia
[13]Japan Agency for Marine-Earth Science and Technology, Yokohama, Japan
[14]MARUM – Center for Marine Environmental Sciences and Department of Geosciences, University of Bremen, Bremen, Germany
[15]Department of Earth and Environmental Sciences, University of Michigan, Ann Arbor, MI 48109, USA
[16]Vrije Universiteit Amsterdam, Faculty of Science, Cluster Earth and Climate, de Boelelaan 1085, 1081HV Amsterdam, the Netherlands
[17]Institute of Numerical Mathematics, Russian Academy of Sciences, Moscow, Russia
[18]Climate and Global Dynamics Laboratory, National Center for Atmospheric Research, Boulder, CO 80305, USA

*Correspondence to*: Masa Kageyama (Masa.Kageyama@lsce.ipsl.fr)

**Abstract.** The Last Glacial Maximum (LGM, ~21,000 years ago) has been a major focus for evaluating how well state-of-the-art climate models simulate climate changes as large as those expected in the future using paleoclimate reconstructions. A new generation of climate models have been used to generate LGM simulations as part of the Palaeoclimate Modelling Intercomparison Project (PMIP) contribution to the Coupled Model Intercomparison Project (CMIP). Here we provide a preliminary analysis and evaluation of the results of these LGM experiments (PMIP4, most of which PMIP4-CMIP6) and compare them with the previous generation of simulations (PMIP3, most of which PMIP3-CMIP5). We show that the global averages of the PMIP4 simulations span a larger range in terms of mean annual surface air temperature and mean annual precipitation, compared to the PMIP3-CMIP5 simulations, with some PMIP4 simulations reaching a globally colder and drier state. However, the multi-model global cooling average is similar for the PMIP4 and PMIP3 ensembles, while the multi-model PMIP4 mean annual precipitation average is drier than the PMIP3 one. There are important differences in both atmospheric and oceanic circulations between the two sets of experiments, with the northern and southern jet streams being more poleward and the changes in the Atlantic Meridional Overturning Circulation being less pronounced in the PMIP4-CMIP6 simulations than in the PMIP3-CMIP5 simulations. Changes in simulated precipitation patterns are influenced by both temperature and circulation changes. Differences in simulated climate between individual models remain large. Therefore, although there are differences in the average behaviour across the two ensembles, the new simulation results are not fundamentally different from the PMIP3-CMIP5 results. Evaluation of large-scale climate features, such as land-sea contrast and polar amplification, confirms that the models capture these well and within the uncertainty of the palaeoclimate reconstructions. Nevertheless, regional climate changes are less well simulated: the models underestimate extratropical cooling, particularly in winter, and precipitation changes. These results point to the utility of using paleoclimate simulations to understand the mechanisms of climate change and evaluate model performance.

## 1 Introduction

The climate of the Last Glacial Maximum (LGM, ~21,000 years ago) has been a focus of the Paleoclimate Modelling Intercomparison Project (PMIP) since its inception. It is the most recent global cold extreme, and as such has been widely documented and used for benchmarking state-of-the-art climate models (Braconnot et al., 2012; Harrison et al., 2014, 2015). The increase in global temperature from the LGM until now (~4° to 6°C, Annan and Hargreaves, 2015; Friedrich et al., 2016) is of the same order of magnitude as the increase projected by 2100 CE under moderate to high-end emission scenarios. The LGM world was very different from the present one, with large ice sheets covering northern North America and Fennoscandia, in addition to the Greenland and Antarctic ice sheets still present today. These additional ice sheets resulted in a lowering of the global sea level by ~120 m, which induced changes in the land-sea distribution. The closure of the Bering Strait and the exposure of the Sunda and Sahul shelves between southeast Asia and the maritime continent are the most prominent of these changes in land-sea geography. Atmospheric greenhouse gases (GHGs) concentrations were lower than pre-industrial (PI) values, leading to cooling in addition to that induced by the large ice sheets. The cooling is more pronounced in the high latitudes than in the tropics, and greater over land than ocean. The polar amplification and the land-sea contrast signals simulated by the previous generation of palaeoclimate simulations (PMIP3-CMIP5) are similar in magnitude (although opposite in sign) to the signals seen in future projections and have been shown to be consistent with climate observations for the historic period and reconstructions for the LGM (Braconnot et al., 2012; Izumi et al., 2013; Harrison et al., 2014, 2015). However, while the models are able to represent the thermodynamic behaviour that gives rise to these large-scale temperature gradients, they underestimate cooling on land, especially winter cooling, and overestimate tropical cooling over the oceans (Harrison et al., 2014). Thus, one question to be addressed with the new PMIP4-CMIP6 simulations is whether there is any improvement in capturing regional temperature changes. The large temperature changes at the LGM make this interval a natural focus for efforts to constrain climate sensitivity, but attempts to do this using the PMIP3-CMIP5 simulations were inconclusive (Schmidt et al., 2014; Harrison et al, 2014), in part because of the limited number of LGM simulations available, and in part because of the limited range of climate sensitivity sampled by these models. Changes in model configuration have resulted in several of the PMIP4-CMIP6 models having substantially higher climate sensitivity than the PMIP3-CMIP5 versions of the same models, and thus the range of climate sensitivity sampled by the PMIP4-CMIP6 models is much wider. This provides an opportunity to re-examine whether the LGM could provide a strong constraint on climate sensitivity (Renoult et al., 2020, Zhu et al., 2021).

The atmospheric general circulation was strongly modified from its modern day conditions by changes in coastlines at low latitudes (DiNezio and Tierney, 2013) and by the presence of the Laurentide and Fennoscandian ice sheets (e.g. Laîné et al., 2009; Lofverstrom et al., 2014, 2016; Ullman et al., 2014; Beghin et al., 2015, Liakka and Lofverstrom, 2018). These changes

in circulation had an impact on precipitation, which was reduced globally (Bartlein et al., 2011) but increased locally, for example in southwestern North American and in the Mediterranean region (e.g., Kirby et al., 2013; Beghin et al., 2016: Goldsmith et al., 2017; Lora et al., 2017; Lora, 2018; Lofverstrom and Lora, 2017; Lofverstrom and Liakka, 2016, Lofverstrom, 2020, Rehfeld et al., 2020). The interplay between temperature-driven and circulation-driven changes in regional precipitation at the LGM represents a test of the ability of state-of-the-art models to simulate precipitation changes under future scenarios, where both thermodynamic (e.g. related to the Clausius-Clapeyron relationship) and dynamic (e.g. related to changes in the position of the storm-tracks and extent of the subtropical anticyclones) effects contribute to changes in the amount and location of precipitation (e.g., Boos, 2012; Scheff and Freirson, 2012; Lora, 2018). Evaluation of the PMIP3-CMIP5 simulations showed that models underestimate the LGM reduction in mean annual precipitation over land (Harrison et al., 2014), reflecting the underestimation of temperature changes in the simulations (Li et al., 2013). This resulted in an underestimation of the observed aridity (precipitation minus evapotranspiration). While the models reproduced circulation-induced changes in precipitation in western North America, they showed no increase in precipitation south of the North American ice sheet and only limited impact on the precipitation of the circum-Mediterranean region (Harrison et al., 2014; Lora, 2018; Morrill et al., 2018). Thus, one question to be addressed with the new PMIP4-CMIP6 simulations is whether there is any improvement in capturing regional precipitation changes. One complication here is that most of the reconstructions used to evaluate the PMIP3-CMIP5 simulations were pollen-based and relied on statistical approaches that do not account for the direct impact of low $CO_2$ on water-use efficiency (Prentice and Harrison, 2009; Gerhardt and Ward, 2010; Bragg et al., 2013; Scheff et al., 2017) and could therefore be dry biased. However, new methods have been developed that account for this effect (Prentice et al., 2017) and thus it is possible to determine whether accounting for the effect of low $CO_2$ resolves model-data mismatches in regional precipitation at the LGM.

The LGM boundary conditions also had a strong impact on ocean circulation, as documented via multiple tracers (e.g. Lynch-Stieglitz et al., 2007, Jaccard and Galbraith, 2011, Böhm et al., 2015), which suggest a shallower North Atlantic Deep Water cell and expanded Antarctic Bottom Water (AABW). Besides, Gebbie (2014) uses a combination of synthesis of multiple tracers measured in sediment cores for the LGM and a global tracer transport model to show that these tracers are compatible with a vertical distribution of NADW and AABW similar to today, but that the core of the NADW water mass shoals by 1000m. None of these proposed reconstructions of glacial circulation is consistent with the PMIP3-CMIP5 model results (Muglia and Schmittner, 2015) which all show a deepening of the Atlantic Meridional Overturning Circulation (AMOC), with NADW reaching the ocean floor in the northern North Atlantic for some models. Previous studies show that this increase in AMOC is related to changes in northern extratropical wind stress due to the presence of the high ice sheets (Oka et al. 2012, Muglia and Schmittner 2015, Klockmann et al. 2016, Sherriff-Tadano et al. 2018, Galbraith and de Lavergne 2019). Thus, the simulation of the AMOC, and ocean circulation in general, at the LGM could be highly sensitive to the ice sheet reconstructions used as boundary conditions (see e.g. Ullman et al., 2014; Beghin et al., 2016). There is still some uncertainty about the height and shape (although not the extent) of the LGM ice sheets, so the protocol for the LGM PMIP4-CMIP6 experiment takes this

uncertainty into account by allowing for alternative ice-sheet configurations (Kageyama et al., 2017) in order to test the sensitivity of LGM climate and ocean circulation to ice sheet configuration. The PMIP4-CMIP6 LGM experimental protocol also includes changes in other forcings, including vegetation changes and changes in atmospheric dust loadings and their uncertainties. Thus, the new PMIP4-CMIP6 simulations provide opportunities to examine the response of the climate system to multiple forcings, to calculate the impact of individual forcings through sensitivity experiments, and to investigate how these forcings combine to produce circulation and climate changes in the marine and terrestrial realms.

In this paper, we present preliminary results from the PMIP4-CMIP6 LGM simulations, compare them to the PMIP3-CMIP5 results (Section 3), and evaluate their realism against a range of climatic reconstructions (Section 4). We focus on temperature and precipitation, extratropical circulation, energy transport and the AMOC.

## 2 Material and methods

### 2.1 PMIP3-CMIP5 and PMIP4-CMIP6 protocols for the LGM simulations

The protocol of the LGM experiments changed between the PMIP3-CMIP5 and PMIP4-CMIP6 phases (Kageyama et al., 2017), partly to accommodate new information about boundary conditions and partly to capitalise on new features of the climate models. The main difference between the PMIP3-CMIP5 and PMIP4-CMIP6 simulations is the specification of the ice sheets. The PMIP3-CMIP5 simulations all used the same ice sheet, which was created as a composite of three separate ice-sheet reconstructions (Abe-Ouchi et al., 2015); the PMIP4-CMIP6 protocol allows modelling groups to use one of three separate ice-sheet reconstructions: the original PMIP3-CMIP5 ice sheet to facilitate comparison with the earlier simulations, ICE-6G_C (Argus et al., 2014, Peltier et al., 2015) and GLAC-1D (L. Tarasov, pers. comm.; Ivanovic et al., 2016). All three reconstructions have similar ice-sheet extent, but the height of the Laurentide, Fennoscandian, and West Antarctica ice sheets differ significantly, by several hundred metres in some places. Comparisons of the simulations made with alternative ice-sheet reconstructions will ultimately allow an assessment of the impact of forcing uncertainties on simulated climates.

### 2.2 PMIP3, PMIP3-CMIP5, PMIP4 and PMIP4-CMIP6 models

The LGM model output analysed here are from the PMIP4-CMIP6 and PMIP3-CMIP5 *lgm* experiments. We use the corresponding *piControl* experiments as a reference, which are termed "PI" throughout the manuscript. Some of the models, although following the PMIP3-CMIP5 or PMIP4-CMIP6 protocols, did not formally take part in CMIP (i.e. have not performed the DECK experiments for CMIP6, or have not performed other experiments than PMIP experiments for CMIP5). These are referred to as "PMIP3" and "PMIP4" models in Table 1. We will refer to the full ensemble of PMIP3-CMIP5 and PMIP3-non-CMIP5 experiments as the PMIP3 ensemble, and similarly for the PMIP4 ensemble. Thirteen PMIP4 LGM simulations are currently available, and slightly more than the eleven LGM simulations in PMIP3 (Table 1). The PMIP3 ensemble includes one model that ran an additional sensitivity test to ice-sheet height (GISS-E2R: Ullman et al., 2014) and one model that ran

simulations with and without dynamic vegetation (MPI-ESM-P: Adloff et al., 2018). The PMIP4-CMIP6 ensemble includes three simulations made with updated versions of the models that contributed to PMIP3-CMIP5, specifically IPSLCM, MIROC and MPI-ESM (Table 1). However, the IPSL simulation for PMIP4 does not use the latest IPSLCM6 version specifically developed for CMIP6 due to the impossibility to run the *lgm* experiment with this version. Most of the models that have run the PMIP4-CMIP6 LGM simulations are general circulation models (GCMs) but iLOVECLIM is an Earth System Model of Intermediate Complexity, which is considerably faster than the GCMs. iLOVECLIM and the HadCM3B-M2.1aD GCM are the only models in the ensemble to have run simulations using different ice sheet reconstructions (both models ran with ICE-6G_C and GLAC-1D, and HadCM3B-M2.1aD also ran with the PMIP3 ice sheet). The LGM simulations were either initialised from a previous LGM simulation or were spun-up from the pre-industrial state. The length of the spin-up therefore varies (Table 1), as does the length of the equilibrium LGM simulation in these preliminary analyses. The INM-CM4-8 results are from the beginning of an *lgm* simulation and the model is not yet fully equilibrated. All other models have run for several millennia. Our preliminary analyses are based on variables available by December 14th, 2020. Although several of the PMIP4-CMIP6 have higher climate sensitivity than the equivalent models in PMIP3-CMIP5, this is not reflected in the ensemble analysed here. In fact, the PMIP4-CMIP6 ensemble, as of December 2020, has lower climate sensitivities than the PMIP3-CMIP5 models (Table 1): equilibrium sensitivities to a $CO_2$ doubling from pre-industrial values range from 2.1 to 3.6°C, (mean: 3.0°C) in the current PMIP4-CMIP6 ensemble, while the range is from 2.1 to 4.7°C (mean: 3.4°C) in the PMIP3-CMIP5 ensemble.

All in all, only a minority of models present in the PMIP3 ensemble ran the PMIP4 simulation, so that the PMIP4 ensemble differs from the PMIP3 one because of the update of these models, but mostly because it gathers new models compared to PMIP3. This adds up to the change in protocol from PMIP3 to PMIP4 to explain differences in model results between these two phases of PMIP.

**2.3 Sources of information on LGM climate**

The PMIP3-CMIP5 model simulations were evaluated against two benchmark data sets: pollen-based reconstructions of seasonal temperature (mean annual temperature – MAT, mean temperature of the coldest month – MTCO, mean temperature of the warmest month – MTWA, and growing season temperature indexed by growing degree days above a baseline of 0°C), mean annual precipitation (MAP) and an index of soil moisture (Bartlein et al., 2011); and a compilation of sea-surface temperature (SST) reconstructions (MARGO, 2009).

In the Bartlein et al. (2011) data set, reconstructions at individual pollen sites were averaged to produce an estimate for a 2 x 2˚ grid; reconstruction uncertainties are estimated as a pooled estimate of the standard errors of the original reconstructions for all sites in each grid cell. Although the Bartlein et al. (2011) has good coverage for some regions, coverage was sparse in the tropics, and there were no reconstructions of LGM climate for Australia. Furthermore, not all of the six climate variables were reconstructed at every site, so statistical comparisons were more robust for some variables than others. The majority of the

reconstructions included in the Bartlein et al. (2011) data set used various sorts of statistical calibrations based on modern day conditions and therefore do not account for the impact that changes in $CO_2$ have on water-use efficiency and hence plant distribution. Although Bartlein et al. (2011) were unable to demonstrate a statistically significant difference between statistical reconstructions and model-based inversions (which in principle account for the $CO_2$ effect on plant distribution), their analysis focused on the mid-Holocene where the $CO_2$ effect is small. There is therefore some concern that the data set may overestimate aridity at the LGM. Reconstructions which incorporate the effect of $CO_2$ are now available for Australia (Prentice et al., 2017). Cleator et al. (2020) have used 3D variational data assimilation techniques with a prior derived from the PMIP3-CMIP5 LGM simulations and the Bartlein et al. (2011) and Prentice et al. (2017) pollen-based reconstructions, and incorporating the Prentice et al. (2017) $CO_2$ correction, to produce a new global reconstruction of terrestrial climate at the LGM. In addition to accounting for potential effects of low $CO_2$ on moisture variables at the LGM, this reconstruction produces coherent estimates of seasonal climate variables at many more points than the original pollen-based reconstructions and also extends the geographic coverage.

Tierney et al. (2020) provide a new synthesis of geochemical SST data ($U^{K'}_{37}$, $TEX_{86}$, Mg/Ca, and $\delta^{18}O$) from the LGM (defined as the period from 19,000 to 23,000 years ago) and the late Holocene (defined as the period from 4,000 years ago to the present) time periods. This compilation builds upon the MARGO (2009) collection of $U^{K'}_{37}$ and Mg/Ca data by including new studies published since MARGO was released, as well as expanding the collection to include $TEX_{86}$ and $\delta^{18}O$ of foraminifera. The Tierney et al. (2020) synthesis excludes microfossil-based SST estimates, on the basis that: 1) these include no-analogue assemblages (Mix et al., 1999); 2) imply warmer-than-present subtropical gyres, an inference that has been questioned (Crowley, 2000; Telford et al., 2013); and 3) lack Bayesian proxy-system models that were required for the data assimilation technique used by Tierney et al. (2020). Tierney et al. (2020) use the data along with a model prior from the isotope-enabled Community Earth System Model 1.2 (CESM, Brady et al., 2019) to produce a full-field data assimilation product. Here we use both the data synthesis and the data assimilation products, labelled 'Tierney2020' and 'Tierney2020DA', respectively. Data from the LGM and late Holocene respectively were calibrated using Bayesian models that fully propagate uncertainties (Tierney & Tingley, 2015; Tierney et al., 2018; Malevich et al., 2019; Tierney et al., 2019), yielding a 1,000 member posterior distribution of SSTs. These data were sorted from low to high along the ensemble dimension, and then random error representative of site-level, downcore uncertainty ($N(0,0.5°C)$ was added back to the matrix. This procedure effectively partitions the error variance; i.e., it assumes that at any given site, absolute uncertainty in SST cancels out in the anomaly calculation, while "relative" uncertainty associated with downcore measurement and non-linearities in the calibration model is preserved. The data were then averaged within a 5° x 5° grid and differenced. The standard deviation associated with each gridpoint is calculated from the differenced ensemble dimension.

In the present work, we also use other available reconstructions, all based on at least part of the initial MARGO (2009) reconstructions at the core sites: all are global reconstructions, obtained from this data set via different methods, as summarised by Paul et al., (2020). These datasets are from:

- Annan and Hargreaves (2013), who use the MARGO (2009) dataset, the Bartlein et al. (2019) reconstructions on the continents, as well as the PMIP2 model output to generate a reconstruction of the sea surface temperatures using multiple linear regression;

- Kurahashi-Nakamura et al. (2017), who use the MARGO (2009) data, benthic $\delta^{18}O$ and $\delta^{13}C$ data as well as the MITgcm in combination with the method-of-Lagrange-multipliers/adjoint method to generate a global reconstruction;

- Paul et al. (2020), who produced the GLOMAP2020 data set based on the floral and faunal assemblages data, as well as various sea ice reconstructions from MARGO (2009), together with an optimal gridding method called DIVA to produce monthly global reconstructions. A caveat given in Paul et al. (2020) about this reconstruction is that it may be too warm by 0.5 to 1.0°C due to impacts of changes in seasonality and in the thermal structure of the ocean that are not taken into account in their reconstructions, as well as the impact due to heterogeneous spatial sampling.

These datasets reflect different approaches and choices of initial datasets (only geochemical data for the Tierney et al. (2020) reconstructions, for which the sites are often close to the coasts, or only floral and faunal assemblages for GLOMAP2020), which yields a range a results with illustrate the uncertainty of the SST reconstructions. A crucial difference between the Tierney et al. (2020) synthesis and the other data sets used here is that the former implies more extensive tropical cooling during the LGM (-2.5˚C, vs -1.5˚C for MARGO, -1.2°C for GLOMAP2020, -1.6°C for Annan and Hargreaves, 2013, -1.7°C for Kurahashi-Nakamura et al., 2017). This can be attributed to the exclusion of the microfossil data as well as recalibration of the $U^{K'}_{37}$ proxy with the BAYSPLINE model (Tierney & Tingley, 2018), which corrects for an observed reduced sensitivity of $U^{K'}_{37}$ to SST above ca. 24˚ C. The data-assimilated product from Tierney et al. (2020) is even cooler, which might be related to the choice of the global model for the assimilation. A further comparison is presented in Paul et al. (2019).

### 2.4. Data-model comparisons

We compare the model simulations to palaeoclimate data, focusing on large-scale features and regional changes. In these comparisons, the reconstructions are expressed as mean values and the uncertainty by the standard error of the reconstructions. Model outputs were extracted only for the grid cells where there are observations. Model uncertainty is represented by the standard deviation of 10,000 averages over 50 years randomly picked in the > 100-year-long timeseries of model outputs. Thus, model uncertainty is not, strictly speaking, equivalent to reconstruction uncertainty but merely provides some measure of the variability engendered by sampling the simulated climate.

### 3 Model results

### 3.1 Temperature

The global and annual mean temperature in the PMIP4 LGM simulations is between 3.3 and 7.2°C cooler than the PI simulations (Fig.1, Table S1). The largest changes in temperature between the LGM and PI simulations (Fig. 2) is found over

the Laurentide and Fennoscandian ice sheets, reflecting the significant changes in surface height and albedo caused by the ice sheets. Colder conditions are registered in the northern mid and high latitudes, partly reflecting the advection of the cold temperature anomalies downwind of the ice sheets. The cooling in the tropics, which results from both the lower atmospheric GHG concentrations and the remote influence of the northern ice sheets, is more muted. As expected, the simulations show larger changes over the land than over ocean. The ratio between the LGM – PI mean surface air temperature anomaly over land and the anomaly over the ocean ranges from 1.0 to 1.6 over the Tropics (30°S to 30°N) and from 1.90 to 5.5 for globally averaged temperatures. Zonally averaged temperatures (Fig. 1) confirm that the PMIP4 ensemble also shows the expected polar amplification of temperature changes in both northern and southern hemispheres.

Although the broadscale patterns of temperature changes are similar, there are differences between the PMIP4 and PMIP3 ensembles. The PMIP4 ensemble average is warmer than the PMIP3 ensemble average (Fig. 2 bottom) over North America, south of the ice sheet, over the Labrador and Nordic Seas and the Tibetan Plateau. On the other hand, the PMIP4 average is colder than the PMIP3 one in regions close to West Antarctica, over some areas of the Laurentide ice sheet, over the marine part of the Fennoscandian ice sheet and in the North Atlantic and the northern part of the North Pacific. The largest difference between the PMIP3 and PMIP4 averages is over the northern North Atlantic and Nordic Seas, probably reflecting differences in sea-ice cover in these areas. Zonally averaged temperatures (Fig. 1) show that the PMIP4 global mean annual temperature LGM – PI anomalies spread over a larger range than the PMIP3 ensemble, with a few PMIP4 models (in particular the three HadCM3 simulations and CESM1.2) which show larger cooling than the coldest PMIP3 models. Nonetheless, the multi-model average of the global mean annual temperature LGM – PI anomalies are similar for both ensembles (-4.71°C for the PMIP3 ensemble, -4.77°C for the PMIP4 ensemble, see supplementary Table S1).

The northern extratropics are slightly colder in the PMIP3 simulations (multi-model LGM – PI MAT anomaly of -9.5°C) than in the PMIP4 simulations (multi-model average of -8.8°C). The minimum and maximum cooling over the PMIP3 and PMIP4 ensembles are also very similar. The PMIP3 and PMIP4 simulations yield similar cooling in the tropics (multi-model average of -2.8°C for the PMIP3 ensemble, and of -2.7°C for the PMIP4 ensemble, with similar minima and maxima, see Table S1). However, the cooling of the southern extratropics is more variable in the PMIP4 simulations (-1.2 to ca -8.15°C) than in the PMIP3 simulations (-2.4 to ca -5.8°C) and its multi-model average is larger for the PMIP4 ensemble (-4.8°C, compared to -2.8°C for the PMIP3 ensemble). Therefore most of the difference in the global average cooling, which ranges from -3.3 to -7.2°C in the PMIP4 simulations and between -2.7 and -5.7°C in the PMIP3 simulations, stems from differences in the simulated temperatures over the southern hemisphere. It is difficult to assign these differences between the PMIP3 and PMIP4 ensembles to a single reason, since both models and protocols have changed between these two phases. Sensitivity experiments and in-depth study of the experiments carried out with the PMIP3 and PMIP4 protocols but with the same models will be necessary to disentangle the reasons for the differences between the PMIP3 and PMIP4 results. It is in fact rather intriguing that the average cooling over the North American ice sheet is larger in the PMIP4 ensemble given both the ICE-6G_C and the GLAC-

1D reconstructions yield significantly lower altitudes than the PMIP3 ice sheet reconstruction, used in all the PMIP3 experiments.

## 3.2 Atmospheric and oceanic circulation

The PMIP4-CMIP6 models simulate large changes in the northern hemisphere upper tropospheric atmospheric circulation (Fig. 3), in response to LGM boundary conditions, in particular over North America and the North Atlantic. The North Atlantic jet stream is narrower and stronger compared to the PI, as shown by an increase reaching more than 10 m/s in the 250 hPa zonal wind south of the Laurentide ice sheet and extending into the North Atlantic, and a decrease in zonal wind to the northwest and southeast of these regions. The strengthening and narrowing of the North Atlantic jet stream was also a characteristic of the PMIP3-CMIP5 simulations (Beghin et al., 2016). However, in the PMIP4-CMIP6 simulations the jet stream extends further north than in the PMIP3 simulations (Fig. 3, bottom), most prominently near the Laurentide ice sheet. This could be because the Laurentide ice sheet is lower in the ICE-6G reconstruction than the ice sheet used in the PMIP3-CMIP5 simulations (see e.g. Ullman et al., 2014; Beghin et al., 2015; Lofverstom et al., 2016) but may also reflect changes in the representation of the zonal winds between the two sets of simulations. This is supported by the fact that there are differences between the PMIP3-CMIP5 and PMIP4 simulations away from the Laurentide ice sheet, in particular over the Southern Ocean, where the jet stream is also located more poleward in the PMIP4 than the PMIP3 simulations. Sensitivity experiments using the PMIP3-CMIP5 ice sheets with PMIP4 models, as planned in the PMIP4 LGM experiment protocol (Kageyama et al., 2017), should help resolve the question of whether differences in model treatment or boundary conditions are responsible for the differences in atmospheric circulation between the two ensembles.

The extent of the North Atlantic Deep Water cell (NADW, identified on Fig. 4 by the depths for the which the Atlantic meridional overturning streamfunction at 30°N is positive) simulated by PMIP4 models is very similar for LGM and PI, except for iLOVECLIM and IPSLCM5A2, which show a very large deepening of the NADW cell for LGM (Fig. 4). Two of the PMIP4-CMIP6 models (INM-CM4-8 and MIROC-ES2L) show a deep NADW cell reaching the ocean floor in the North Atlantic, whereas five of the PMIP4-CMIP6 models (MPI-ESM1.2, UoT-CCSM4, AWIESM2, CESM1.2, HadCM3) simulate a clear Antarctic Bottom Water (AABW) in the North Atlantic. UoT-CCSM4 and CESM1.2 even shows a shallowing of the NADW cell for LGM. The intrusion of AABW cell (defined by negative values in the Atlantic meridional overturing streamfunction at 30°N) into the North Atlantic was shown by some of the PMIP3-CMIP5 simulations (CCSM4, MPI-ESM-1.0P), but not as much as the PMIP4 simulations (AWIESM2, CESM1.2, MPI-ESM-1.2, UoT-CCSM4, and the three HadCM3 simulations, Fig. 4 and Muglia et al., 2015). Five of the PMIP3-CMIP5 models produced a NADW cell reaching the ocean floor in the North Atlantic, and only two had extensive AABW. The maximum strength of the NADW cell itself strengthens in all of the PMIP4 simulations, by as much as 11 Sv for IPSLCM5A2. This strengthening is consistent with PMIP3-CMIP5 results and is likely to be associated with the vigorous surface wind over the northern North Atlantic (Muglia and Schmittner 2015, Sherriff-Tadano and Abe-Ouchi 2020), and the closure of the Bering Strait (Hu et al. 2015). The strength of the AMOC

reduces south of 30˚N in UoT-CCSM4 (see Supplementary Figure S2). iLOVECLIM performed simulations of LGM with two different ice sheet reconstructions (ICE6G, GLAC1D) and show a weaker NADW cell in GLAC-1D than that produced by ICE-6G_C (Fig. 4). This weakening is likely to be associated with a lower topography of the ice sheet of GLAC1D (e.g. Zhang et al. 2014). On the other hand, HadCM3 was used with the PMIP3, ICE-6G_C and GLAC-1D ice sheets, and the results in terms of AMOC are very similar for the ICE-6G_C and GLAC-1D ice sheets, for which the AMOC slightly strengthens compared to PI, while the AMOC is similar to the PI one for the simulation using the PMIP3 ice sheet.

These circulation changes in the Atlantic Ocean are reflected in the total ocean heat transport (Fig. 5, bottom, the PMIP4 results available for this analysis are from all simulations but the HadCM3 simulations). MPI-ESM1.2 simulates an increase in northward ocean heat transport at all latitudes for the LGM compared to PI, while MIROC-ES2L simulates an increase in this transport from 15°S to 60°N. UoT-CCSM4 and CESM1.2 are the only models simulating a decrease in northward heat transport over a significant range of latitudes, from 50°S to 70°N, in the *lgm* run compared to the *piControl* one. INCM4-CM4-8 simulates an increased ocean transport south of 20°N. IPSLCM5A2's ocean transport decreases south of 30°S and between the equator and 30°N but significantly increases in the southern tropics. All PMIP4 models simulate an increase in northward atmospheric heat transport, in the tropics and up to 50°N, in the *lgm* simulation compared to *piControl*. MIROC-ES2L simulates an increase up to 70°N (Fig. 5, middle). In summary, all models simulate an increase, in their *lgm* run compared to *piControl,* in northward heat transport (Fig. 5, top) in the tropics and northern mid-latitudes, although in the UoT-CCSM4 and CESM1.2 models the increase is confined between ~10 and 50°N. This increase in northward heat transport in the tropics and northern mid-latitudes during the LGM as compared to PI was also simulated by most PMIP3-CMIP5 models. Given that the magnitude of the heat transport increase is similar in the PMIP4 and PMIP3-CMIP6 simulations, the warmer temperatures at high northern latitudes in the PMIP4-CMIP6 simulations cannot be due to differences in northward ocean heat transport.

### 3.3 Hydrological cycle

The large-scale gradients in precipitation are similar in the mutli-model average of the PMIP4 LGM and PI simulations (Fig. 6, top left), with maximum precipitation in the tropics (Intertropical Convergence Zone and monsoon regions) and secondary maxima in the mid-latitudes, corresponding to the position of the North Pacific, North Atlantic, and Southern Ocean storm-tracks. The PMIP4 models show a decrease in precipitation between the LGM and PI in all these high precipitation areas (Fig. 6, bottom left and Fig. 7, top left). There are some regions where precipitation increases at the LGM compared to the PI: at least 9 PMIP4 models (as shown by the areas which are not stippled) show more precipitation over the subtropical Pacific Ocean and to the south of the Laurentide ice sheet, over southern Africa and over the Iberian Peninsula, and some simulate an increase in precipitation over the northern and southern subtropical zones in the Pacific and over the southern subtropical zone in the Atlantic. However, the areas with decreased precipitation are much more extensive than areas with increased

precipitation, so zonal averages for the southern extratropics, tropics and northern extratropics (Fig. 7) all show a decrease in precipitation.


The broadscale patterns of change in precipitation in the PMIP4 simulations are similar to those found in the PMIP3-CMIP5 simulations (Fig. 7, top left). However, the PMIP4 multi-model average is drier than the PMIP3-CMIP5 one (Fig. 6), at the global scale as well as for the southern extratropics and for the tropics. It is similar for both ensembles for the northern extratropics. The geographic patterning in the precipitation changes between the PMIP4 and PMIP3-CMIP5 ensembles (Fig.

6, top right) is complex, particularly in the tropical where the wetter-drier-wetter pattern in the meridional direction suggests differences in ITCZ representation between the two generations of models. This is confirmed by the same figure drawn for the PI (Fig. 6, bottow right), which shows very similar patterns in the PMIP3 vs PMIP4 anomalies. Both ensembles show a consistent decrease in zonally averaged precipitation in the southern and northern extratropics (Fig. 7). As for the mean annual temperature, the simulated range of precipitation changes is larger for PMIP4 ensemble compared to the PMIP3 one, except

for the northern extratropics for which both ensembles show a similar range (Fig. 7 and Fig. 2).

Evapotranspiration patterns in the PMIP4 LGM and PI simulations are characterized by maximum values in the subtropics and decrease towards high latitudes. The models simulate a global decrease in LGM evapotranspiration relative to the PI that strongly peaks over and around the northern hemisphere ice sheets (Fig. 8, left). These results are in agreement with the broad

patterns of the PMIP3-CMIP5 ensemble, except for a stronger decrease in evaporation in the northern North Atlantic, which corresponds to the larger average cooling in these regions in the PMIP4 ensemble, compared to the PMIP3 ensemble. As a result, net precipitation (precipitation minus evapotranspiration) in the PMIP4 ensemble is higher at the LGM than the PI in the extratropics—particularly over the mid-latitude eastern Pacific in both hemispheres and over most of North America— with the exception of the North Atlantic where evaporation decreases are more localized and do not compensate for the

reductions in precipitation (Fig. 8, right). This, together with colder temperatures, could help explain why the PMIP4 models simulate a stronger AMOC at the LGM. Substantial reductions in continental net precipitation only occur over tropical South America and high-latitude regions, over the Labrador Sea and its surrounding ice sheets, while Africa, Australia, and the mid-latitude regions of Eurasia and the Americas see little change or even increased net precipitation.

**4 Data-model comparisons**

The evaluation of the PMIP3-CMIP5 LGM simulations showed that large scale climate features, such as the ratio of changes in land-sea temperature, in high-latitude temperature amplification, and in precipitation scaling with temperature were broadly consistent with modern observations (Braconnot et al., 2012; Izumi et al., 2013; Harrison et al., 2014, 2015).

All PMIP3 and PMIP4 models simulate larger cooling over land than over oceans, on average for the tropics and for the globe. Fig. 10 shows averages of model output sampled at sites for which there are reconstructions, compared to the averages of the reconstructed values. Since the different reconstructions do not cover the same sites, the averages of the model values at reconstruction sites differ slightly for each data set. However, for all data sets, the multi-model relationship between the average cooling over land and that over the ocean is approximately linear. Fig. 10 allows a comparison between model output and

reconstructions averaged over land and over oceans, as well as a comparison of the ratio of the land cooling over the ocean cooling. Although the Cleator et al. (2020) data set has a larger spatial coverage than the Bartlein et al. (2011) data set, there is no significant difference between the two data sets for most of the temperature variables across common grid cells (Fig 9). However, the new reconstructions have a reduced range at the warm end, especially between 0 and 40°N, so that for the averages over the tropics, most simulations are recorded as within or warmer than the land-based reconstructions, while they

are within or colder than the Bartlein et al. (2011) reconstructions (Fig. 10, l.h.s). The results for the global averages are fairly consistent for both land-based reconstructions (Fig. 10, r.h.s) but the uncertainty is smaller for the Cleator et al. (2020) data set. All in all, there are as many simulations within the range of globally averaged reconstructed temperatures of Cleator et al. (2020) as that of Bartlein et al. (2011) but the models outside this range tend to be on the warm side for the Cleator et al. (2020) dataset, and both on the warm and cold side for the Bartlein et al. (2011) dataset.


The reconstructions of the LGM - Late Holocene SST anomalies provided by Tierney et al (2020) are colder than the MARGO reconstructions in the tropics and, although this removes the apparent cold-bias shown by some simulations, this results in some simulations falling outside the window of reconstructed SSTs at the warm end. This is even more the case if we compare the results to the data-assimilated product from Tierney et al. (2020), which has a global coverage (Fig. 10, l.h.s, bottom line).

The results from this latter dataset contrasts with the results from other global products, as shown in Supplementary Figure S3. These other global datasets (Annan and Hargreaves, 2013, Kurahashi-Nakamura et al., 2017, GLOMAP2020, Paul et al., 2020) are all derived, at least in part, from the MARGO (2009) dataset, which might explain their overall consistency with the averages estimated by MARGO (2009). We have added the warmest estimate (GLOMAP2020) which includes uncertainties on Fig. 10 to obtain a more complete view of the available reconstructions as of 2020, keeping in mind that the authors of the

GLOMAP2020 reconstruction estimate that it could be biased by 0.5 to 1.0°C in the tropics. The simulated mean annual surface air temperature decreases over the tropical oceans stand between these two extremes. This illustrates that model-related uncertainties are comparable with the uncertainties raising from the multiple approaches taken to reconstruct both the continental and oceanic temperatures.

The ratio for the land-sea difference in changes in mean annual temperature in the tropics in the PMIP4 simulations is compatible with the ratio reconstructed from the Bartlein et al. (2011) and MARGO (2009) data sets. This is also the case if we consider the more recent reconstructions by Cleator et al. (2020) and Tierney et al. (2020), although the multi-model land-sea ratio appear to be smaller than that suggested by the reconstructions. This is the case for both the tropical and global

averages. However, it would not be compatible with a land-sea contrast based on the Cleator et al (2020) dataset and the
GLOMAP2020 dataset, even if the warm bias pointed by its authors is taken into account. We are therefore left with large
uncertainties on the topic of LGM cooling over land and oceans, from the reconstructions as well as from the models. The
uncertainties based on the ensemble of model results and on the ensemble of continental and marine reconstructions are actually
very similar.

The amplification of temperature changes at high northern latitudes compared to the tropics is apparent over both the land and
the ocean domains, although the amplification appears to be smaller in the new data syntheses (Fig. 11), except of the Tierney
et al. (2020) data assimilated product. For the ocean domain, this could reflect the influence of seasonal production on the
extratropical sites, with indicators being more sensitive to summer changes, or to changes in the seasonal production cycle.
Comparisons of the amplification over land areas with the Bartlein et al. (2011) data set suggest that the simulated tropical
cooling is too large in the PMIP3-CMIP5 simulations whereas the extratropical cooling was both larger and smaller than
suggested by the reconstructions in both ensembles. Simulated tropical temperatures are more consistent with or warmer than
the Cleator et al. (2020) reconstructions, suggesting that the apparent over-estimation of tropical cooling in the PMIP3-CMIP5
simulations over land may reflect the paucity of tropical data points in Bartlein et al (2011). However, the discrepancies
between the simulated and reconstructed extratropical land temperatures are still present: there are several PMIP3 and PMIP4
simulations that are much colder than the reconstructions, and many which are warmer than the reconstructions. Although
polar amplification is more muted over the ocean domain, the comparisons show a similar picture to the land-based
comparisons. Simulated tropical ocean temperatures are more compatible with the Tierney et al. (2020) than the MARGO
(2009) synthesis. Simulated extratropical temperature changes in the PMIP3-CMIP5 ensemble are considerably colder than
shown by either of these syntheses, but most tend to be on the warm side of the Tierney et al. (2020) data assimilated product.

The LGM climate is characterised by an increase in temperature seasonality in extratropical regions, with larger changes in
winter than in summer (Izumi et al., 2013). This is confirmed by the Cleator et al. (2020) reconstructions. In general, this
change in seasonality is reproduced by the models, although the ranges of PMIP4 results for winter are less distinct from their
summer counterparts than for the PMIP3 models. The multi-model average seasonality is, however, increased for both
ensembles. The simulated cooling in winter temperature is smaller than indicated by the Bartlein et al. (2011) reconstructions
(Fig. 12, top line). This is not the case compared to the Cleator et al. (2020) reconstructions, with which more models are in
agreement, except for Western Europe which remains a region of model-data discrepancy. The magnitude of the summer
cooling is more consistent between the PMIP4 simulations and the Cleator et al. (2020) reconstructions than between the
PMIP3 simulations and the Bartlein et al. (2011) reconstructions in North America, Europe and extratropical Eurasia. Finally,
the North Atlantic mean annual cooling simulated by the PMIP4 models span a larger range that the PMIP3 ensemble. While
the PMIP3 ensemble showed temperatures within or above the reconstructed ranges from all oceanic datastes, the PMIP4
ensemble average stands within the range of the reconstructions, with six individual models being within this range, 4 above

and 3 below. It is therefore quite difficult to determine the cause of the discrepancy in western Europe winter temperatures, which was previously assigned to an underestimation of the North Atlantic cooling. Some of the PMIP4 simulations are in fact much colder over both the North Atlantic and western Europe, and could be studied to further disentangle this model-data disagreement.

Regional changes in the tropics (Fig. 12, bottom line) are more muted than those in the northern extratropics, and seasonality differences are small. We therefore base our comparisons on the mean annual temperature and mean annual precipitation. Both PMIP3 and PMIP4 multi-model averages underestimate MAT cooling over tropical America, which is consistent for both reconstructions. More PMIP4 results stand within the reconstructed range over tropical America. Over tropical Africa, the PMIP3 models were broadly consistent with the Barltein et al. (2011) reconstructed MAP anomaly but underestimate this cooling if we refer to the more recent Cleator et al. (2020) reconstructions. This is also the case for the PMIP4 models, but 4 simulations (IPSLCM5A2, and the three HadCM3 simulations) are now within the reconstructed range. This is probably related to the simulated tropical SSTs being colder in these simulations. The reconstructed changes in tropical precipitation over America are larger in the Cleator et al. (2020) dataset than in Bartlein et al. (2011) and both PMIP3 and PMIP4 models underestimate the reconstructed drying (Fig. 12). The PMIP4 models, however, all simulate the correct negative sign of the reconstructed precipitation change. There is a large difference between the estimates of precipitation change given by the Bartlein et al. (2011) and the Cleator et al. (2019) data sets for tropical Africa, with the Cleator et al. (2020) reconstructions reducing the drying reconstructed by Bartlein et al. (2011). The ranges of PMIP3 and PMIP4 results are broadly similar over this region, and there are the same number of models (four) within the reconstructed range of Cleator et al. (2020), while no model result was compatible with the Bartlein et al (2011) reconstructions. All other models underestimate the change or even simulate an increase in precipitation. All in all, the simulated changes in precipitation are therefore more consistent with the newer data set. Thus, there is no systematic improvement in the simulation of tropical climates between the PMIP4 and PMIP3 ensembles.

The six data syntheses can be used to try and constrain the global MAT change from LGM to PI. There is a good correlation between the change in global average MAT over the reconstruction grid points and computed taking all the model grid points into account (Fig. 13). The idea here is therefore to take advantage of this relationship to obtain a range in the global MAT anomaly from the reconstructions.There are models with results below, within and above the average of all of the reconstructions except MARGO (2009) and GLOMAP2020 for which no model simulates MAT LGM – PI anomalies above the reconstructed range of values. Retaining only the models which produce changes in MAT consistent with the reconstructions (and reconstruction uncertainty), the globally averaged change in MAT is between -5.7 and -3.7°C using the Bartlein et al. (2011), between -6.7 and -4.6°C using the Cleator et al. (2020) data sets, between -4.9 and -3.2°C for the Tierney et al. (2020) data set, and above -3.9°C and -4.4°C for the MARGO (2009) and GLOMAP2020 data sets, respectively. Taken

altogether, these estimates are consistent with previous estimates, which indicate changes in MAT of between 4 and 6°C (Annan and Hargreaves, 2015; Friedrich et al., 2016).

## 5 Conclusions and perspectives

The results from the PMIP4 models differ from those of the PMIP3 ensemble in several ways. The multi-model global cooling is similar in both ensembles but the PMIP4 ensemble range is larger, with 4 simulations showing colder results than the coldest PMIP3 model. This feature mainly arises from the southern hemisphere extratropics and it is currently difficult to disentangle whether it is due to the PMIP4 model ensemble being largely different from the PMIP3 ensemble, or to the changes in protocol from PMIP3 to PMIP4 (see also Zhu et al., 2021). The change in the ice sheets appears to have an impact on atmospheric

circulation over North America and the North Atlantic. The AMOC increases less in the PMIP4 than in the PMIP3 simulations, and the depth of the NADW cell remains more stable, except for two models, in contrast with more than half the models of the PMIP3-CMIP5 ensemble, which simulated a large deepening of this cell. This could be due to the changes in atmospheric circulation over the North Atlantic, as well as changes in the North Atlantic freshwater balance. Changes in precipitation are generally similar for the PMIP3 and PMIP4 ensembles and characterised by less precipitation overall. Reduced evaporation

due to colder temperatures partially compensates for the reduction in precipitation, so that areas of negative and significant LGM - PI anomalies in net precipitation (i.e. precipitation minus evaporation) are larger than areas with positive LGM - PI precipitation anomalies. However, both precipitation and net precipitation changes show large spatial heterogeneity, and different regional-scale patterns of change between the PMIP4 and PMIP3 ensembles, which appears to be related to the performance of the model ensemble for PI. Additional sensitivity experiments are needed to separate the effects of changes in

model configuration and sensitivity on general circulation features, such as the position of the jet streams, from the effects of differences in boundary conditions, such as the improved realism of the ice sheet configuration.

The PMIP4-CMIP6 ensemble confirms that the models simulate large-scale thermodynamic behaviour common to historical and future simulations, such as land-sea contrast and polar amplification. The results from the PMIP3 and PMIP4 align on the

475 same relationships for these large-scale characteristics of climate change. The new reconstructions of Tierney et al. (2020) and Cleator et al. (2020) are in better agreement than with the reconstructions from Bartlein et al. (2011) and MARGO (2009) used to evaluate these features previously (Braconnot et al., 2012; Izumi et al., 2013; Harrison et al., 2014, 2015), for the tropical and global averages. However, global reconstructions of the surface ocean temperatures, such as the Tierney et al. (2020) data assimilated product, the GLOMAP2020 data by Paul et al. (2020) and the reconstructions by Annan and Hargreaves (2013)

and Kurahashi-Nakamura et al. (2017) show a wide range of results in terms of tropical temperatures (from -1 to -4°C), which prevents firm conclusions on the model-data comparisons based on such global reconstructions. Interestingly, evaluating the uncertainty on tropical cooling from the PMIP model ensemble, on the hand hand, and from the ensemble of continental and marine reconstructions on the other yields very similar results.

The simulated global change in MAT averaged over all the grid cells where reconstructions are available is well correlated with the global average on all model grid points, providing a constraint on the value of the global LGM cooling compared to PI. Using the terrestrial data sets as a constraint, indicates a global cooling between -6.7 and -3.7°C, while using Tierney et al. (2020) as a constraint indicates a global cooling of -4.9 to -3.2°C and using the MARGO (2009) and GLOMAP2020 datasets constrain the global average to be above -3.9°C and -4.4°C, respectively. The constrained range (-6.7 to -3.2°C) is larger than previous estimates (-4 to -6°C).

There is no obvious improvement in model performance at a regional scale between the PMIP3 and PMIP4 ensembles. In some cases (e.g summer temperature over western Europe and extratropical Asia), the PMIP4 ensemble demonstrates a better ability to capture the changes depicted by the reconstructions, in some other (e.g. winter temperatures over Europe, mean annual precipitation over tropical America), the PMIP4 ensemble is still far from the reconstructed values.

Our analyses present a first picture of the PMIP4 LGM experiments. Results from CMIP6 models with high climate sensitivity have only recently become available (Zhu et al., 2021), and will need to be considered in a full assessment of the PMIP4-CMIP6 simulations. Sensitivity experiments, for example to different ice sheet configurations, are needed to disentangle the impact of model improvements from those related to using more realistic boundary conditions. Additional planned simulations will also help to disentangle the impacts of changes in vegetation and aerosol loading on the LGM climate. A more systematic evaluation of the simulated climates, using a wider range of palaeoenvironmental data, will be helpful in understanding why there are persistent mismatches between the simulations and reconstructions at a regional scale. Nevertheless, this preliminary analysis demonstrates the utility of the PMIP4-CMIP6 simulations in addressing questions about the response of climate to large changes in forcing and illustrates the need to investigate the causes of inter-model differences in these responses.

**Author contribution:**

MK, MLK, UM, SST, TV, AAO, NB, DC, LJG, RFI, KI, ALG, FL, GL, PAM, RO, WRP, CJP, AQ, DMR, XS, PJV, EV, JZ provided the model output presented in this manuscript. SPH, AP, JET provided new climatic reconstructions. MK, SPH, MLK, ML, JML, SST, TV, AP prepared the figures and wrote the manuscript.

**Acknowledgements**

The authors would like to thank all the modelling groups who provided the PMIP3 and PMIP4 output for this analysis, WCRP, CMIP panel, PCMDI, ESFG infrastructures for sharing data, WCRP and CLIVAR for supporting the PMIP project. M.K. acknowledges the use of the IPSL (ESPRI – Ensemble de Services Pour la Recherche l'IPSL – computing and data centre (https://mesocentre.ipsl.fr) which is supported by CNRS, Sorbonne Université, Ecole Polytechnique, CNES and through

national and international grants) and LSCE storage and computing facilities for the analyses presented in this manuscript. The IPSL model was run on the Très Grande Infrastructure de Calcul (TGCC) at Commissariat à l'Energie Atomique (gen2212 project). XS and GL acknowledge the German Climate Computing Center (DKRZ) for the AWI-ESM simulations. We thank the two reviewers and editor for their helpful comments on the first version of this manuscript.


## Code and data availability

The data shown in this manuscript can be found on the IPSL repository (doi to be inserted here). The results from the PMIP3-CMIP5 and PMIP4-CMIP6 models can be found on the ESGF (Earth System Grid Federation).

## Competing interests

The authors declare no competing interests.

## Financial support

M. K. is funded by CNRS. SPH acknowledges funding from the ERC-funded project GC2.0 (Global Change 2.0: Unlocking the past for a clearer future, grant number 694481). The MPI-M contribution was supported by the German Federal Ministry of Education and Research (BMBF) as a Research for Sustainability initiative (FONA) through the project PalMod (FKZ: 01LP1504C). JET acknowledges funding from the US National Science Foundation (AGS-1602301) and the Heising-Simons Foundation (#2016-015). RFI acknowledges partial support from the UK Natural Environment Research Council (#NE/K008536/1), and additionally from UK Research and Innovation (#MR/S016961/1) who also support LJG. CJP acknowledges funding from the US National Science Foundation (AGS-2002397) and the Heising-Simons Foundation (#2016-012). XS and GL acknowledge the support of the PACMEDY and PALMOD2 projects. Ayako Abe-Ouchi acknowledges the financial support from JSPS KAKENHI grant 17H06104 and MEXT KAKENHI grant 17H06323, and the support from JAMSTEC for the use of the Earth Simulator supercomputer.

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

**Table 1: PMIP3 and PMIP4 models analysed in the present study. The spin-up duration is only given for the new PMIP4-CMIP6 models.**

| Model | Climate sensitivity $\Delta T^{eq}$ | Reference | Ice sheets | Spin-up duration (years) | PMIP/CMIP phase and rip(f) | Additional comments |
|---|---|---|---|---|---|---|
| CCSM4 | 2.9 | Brady et al., 2013 | PMIP3 | | PMIP3-CMIP5 r1i1p1 | |
| CNRM-CM5 | 3.3 | Voldoire et al., 2013 | PMIP3 | | PMIP3-CMIP5 r1i1p1 | |
| COSMOS-ASO | 4.1 | Raddatz and others (2007); Budich and others (2010) Wetzel and others (2010) | PMIP3 | | PMIP3 r1i1p1 | |
| FGOALS-g2 | 4.4 | Zheng and Yu (2013) | PMIP3 | | PMIP3-CMIP5 | |
| GISS-E2-R | 2.1 | Ullman et al. 2014 | PMIP3 | | PMIP3-CMIP5 r1i1p150 | PMIP3 ice sheet |
| GISS-E2-R | 2.1 | Ullman et al. 2014 | PMIP3 | | PMIP3-CMIP5 r1i1p151 | ICE-5G ice extent but lower Laurentide Ice Sheet altitude |
| IPSL-CM5A-LR | 4.1 | Dufresne et al., 2013 | PMIP3 | | PMIP3-CMIP5 r1i1p1 | |
| MIROC-ESM | 4.7 | Sueyoshi et al. 2013 | PMIP3 | | PMIP3-CMIP5 r1i1p1 | initial ocean state was taken from PMIP2 MIROC4m |
| MPI-ESM-P | 3.5 | | PMIP3 | | PMIP3-CMIP5 r1i1p1 | AO, initial state for spin-up from PMIP2 simulation |
| MPI-ESM-P | 3.5 | Adloff et al., 2018 | PMIP3 | | PMIP3-CMIP5 r1i1p2 | AOV |

| MRI-CGCM3 | 2.6 | | PMIP3 | | PMIP3-CMIP5 r1i1p1 | |
|---|---|---|---|---|---|---|
| AWI-ESM1-1-LR (shortname: AWIESM1) | 3.6 | Sidorenko et al., 2015; Lohmann et al., 2020 | ICE-6G_C | 1300 | PMIP4-CMIP6 | |
| AWI-ESM-2-1-LR (shortname AWIESM2) | 3.6 | Sidorenko et al., 2019 | ICE-6G_C | 600 | PMIP4-CMIP6 | |
| CESM1.2 | 3.6 | Tierney et al., 2020 | ICE-6GC | 1800 | PMIP4-CMIP6 | |
| UoT-CCSM4 | 3.2 | Peltier and Vettoretti, 2014 Chandan and Peltier, 2018, Chandan and Peltier, 2017 | ICE-6G_C | 2900 years | PMIP4-CMIP6 | |
| HadCM3B-M2.1aD | 2.7 | Valdes et al. 2017 | GLAC-1D ICE-6G_C PMIP3 | 400 400 2900 | PMIP4-CMIP6 | All simulations were initialised from a long (>5000 year) LGM run that used the same model configuration, but ICE5G boundary conditions (ice mask, global orography, bathymetry, land-sea mask) and PMIP3 trace gases. The climatologies were calculated from the one hundred years following the spin-up period. |
| iLOVECLIM1.1.4 | 3.2 (after 2500 years) | Lhardy et al., 2020 | GLAC-1D ICE-6G_C | 5000 | PMIP4 | 5000 years from a PI restart; EMIC |
| iLOVECLIM1.1.4 | 3.2 (after 2500 years) | Lhardy et al., 2020 | ICE-6G_C | 5000 | PMIP4 | 5000 years from a PI restart; EMIC |
| INM-CM4-8 | 2.1 | Volodin et al., 2018 | ICE-6G_C | 50 | PMIP4-CMIP6 r1i1p1f1 | |
| IPSLCM5A2 | | Sepulchre et al., 2020 | ICE-6G_C | 1200 | PMIP4-CMIP6 | Spin-up from piControl |

| MIROC-ES2L | 2.7 | Ohgaito et al., 2020<br>Hajima et al., 2020 | ICE-6G_C | 8960 | PMIP4-CMIP6<br>r1i1p1f2 | First 6760 years integrated using the MIROC-ES2L physical core; following 2200 years integrated using MIROC-ES2L |
| MPI-ESM1.2 | 2.77 | Mauritsen et al., 2019 | ICE-6G_c | 3850 | PMIP4-CMIP6<br>r1i1p1f1 | 3850 years after restart from a previous lgm simulation. |

**Figure 1: Mean annual surface air temperatures LGM – PI anomalies in °C. a) zonal means, PMIP3 model results shown as dashed lines, PMIP4 model results shown as thick solid lines; b) global means, PMIP3 model results shown by crosses, PMIP4 models shown by filled circles; averages over c) the southern extratropics (90°S to 30°S), d) the tropics (30°S to 30°N) and e) the northern extratropics (30°N to 90°N).**

**Figure 2: LGM mean annual temperature (in °C) simulated by the ensemble of PMIP4 models (top), LGM – PI mean annual temperature anomaly (in °C) simulated by the same models (middle, where stippling shows where models do not agree on the sign of changes), difference between the PMIP4 and PMIP3 ensembles (in °C, bottom). The PMIP4 average is based on models listed in Table 1, except for iLOVECLIM simulations, which are at lower resolution. The PMIP3 average is based on all PMIP3 models, except the GISS-E2-p151 simulation which did not use the PMIP3 ice sheet for its boundary conditions.**

**Figure 3: Same as Figure 1 but for the 250 hPa zonal wind. The PMIP4 average is based on all models listed in Table 1. The PMIP3 average is based on all PMIP3 models in Table 1, except the GISS-E2-p151 simulation which did not use the PMIP3 ice sheet for its boundary conditions.**

**Figure 4: Mean Atlantic Meridional Overturning Circulation (mean meridional stream function for the Atlantic Ocean at 30°N) simulated by the PMIP3 and PMIP4 models for PI and LGM. Numbers in Sv indicate the LGM – PI anomaly in terms of maximum**

**Atlantic meridional overturning streamfunction. Numbers in m indicate the LGM – PI anomaly in terms of NADW vertical extension, the NADW vertical extent being defined here as the depths over which the mean meridional stream function for the Atlantic Ocean at 30°N is positive.**

**Figure 5: Meridional energy transport for the PI reference state (l.h.s) and LGM - PI anomaly (r.h.s). Top: total energy transport, middle: atmospheric energy transport, bottom: oceanic energy transport.**

**Figure 6: Top left: PMIP4-CMIP6 multi-model LGM mean annual precipitation, in mm/day. Bottom left: PMIP4-CMIP6 multi-model LGM – PI mean annual precipitation anomaly (mm/day), with stippling showing areas where less than 9 models agree on the sign of change. Top right: Difference between the PMIP4-CMIP6 and the PMIP3 multi-model means of the LGM mean annual precipitation (mm/day). Bottom right: Difference between the PMIP4-CMIP6 and the PMIP3 multi-model means of the PI mean annual precipitation (mm/day).**

**Figure 7: Same as Figure 2 for mean annual precipitation, in mm/year.**

**Figure 8: same as Fig. 1 for mean annual evaporation (l.h.s) and mean annual net precipitation (precipitation - evaporation, r.h.s). All values are in mm/day. Stippling shows areas where less than 9 models agree on the sign of change.**

**Figure 9: Comparison of terrestrial climate variables from the combined Bartlein et al. (2011) and Prentice et al. (2017) data set and from the Cleator et al. (2020) reconstruction using data assimilation, averaged over 20° latitudinal bands. The variables are mean**

**annual temperature (MAT), mean temperature of the coldest month (MTCO), mean temperature of the warmest month (MTWA), and mean annual precipitation (MAP). The orange boxplots show the results from the Bartlein et al. (2011) and Prentice et al. (2017)**

combined data set, the dark blue boxplots for the reconstructions by Cleator et al. (2020) at sites for which there are reconstructions in the combined dataset, and the green boxplots show the results for the full reconstructions from Cleator et al. (2020).

**Figure 10:** LGM - PI mean annual temperature anomaly over land vs. LGM - PI mean annual temperature anomaly over oceans, averaged over the tropics (30°S-30°N, l.h.s) and over the globe (r.h.s). The model output considered for the averages is taken only on grid points for which there are reconstructions. The top plots are based on the reconstructions used to evaluate the PMIP3-CMIP5 models: the Bartlein et al. (2019) database and the MARGO (2019) SST reconstructions. The bottom plots are based on the most recent reconstructions: Cleator et al. (2019) for terrestrial data and Tierney et al. (2019) for the SSTs.

**Figure 11:** LGM - PI mean annual temperature anomaly over the northern extratropics (30-90°N) vs. over the northern tropics (0-30°N). The model output considered for the averages is taken only on grid points for which there are reconstructions. The four panels are based on the data syntheses of Bartlein et al. (2011) (top left), MARGO (2009) (top right), Cleator et al. (2019), Tierney et al. (2019).

**Figure 12:** Data-model comparisons for North America (20-50°N, 140-60°W), the North Atlantic Ocean (30-50°N, 60°W-10°W), Western Europe (35-70°N, 10°W-30°E) , Extratropical Asia (35-75°N), Tropical Americas (30°S-30°N, 120-60°W), Africa (35°S-35°N, 10°W-50°E), and tropical oceans (30°S-30°N). MTCO: Mean Temperature of the Coldest Month, MTWA: Mean Temperature of the Warmest Month, MAT: Mean Annual Temperature, MAP: Mean Annual Precipitation, MATocean: Mean Annual Temperature over the oceans  The error bars for the reconstructions are based on the standard error given at each site: the average and associated standard deviation over the specific area are obtained by computing 10000 times the average of randomly drawn values in the gaussian distributions defined at each site by the reconstruction mean and standard error, taken as the standard deviation of the gaussian. Uncertainty for the model results have been computed based on the 10000 randomly picked groups of 50 years which were averaged to obtain 10000 estimates of the 50-year average for a specific region and variable. These were so small they do not appear on the plots.

**Figure 13:** Relationships between global mean temperature changes (x-axes, average computed on all model points) and global mean temperature changes for grid points where there are reconstructions (y-axes, 1 plot per data set). For each plot/data set, the models whose average falls in the range of the average of the reconstructions are marked by vertical dotted lines down to the x-axis.