# Peer review of "The PMIP4 Last Glacial Maximum experiments: preliminary results and comparison with the PMIP3 simulations"

_Climate of the Past, 2019_

## Referee Comment (RC1) · Anonymous Referee #1 · 20 May 2020

First of all, I apologise to the authors for my late review.

The article presents preliminary results of the new PMIP4-CMIP6 exercise, compare it to the former PMIP3-CMIP5 simulation ensemble, along with new LGM temperature databases for continental and oceanic temperatures. It is an important and interesting update of the LGM experiments, well organized, and provides a state-of the art on recent advances in climate modeling and data compilation. It is very descriptive but I enjoyed very much reading it. Sometimes, figures are not easy to read, but I don't see how it could be improved given all the dimensions the model/data comparison requests. I recommend publication with some minor rectifications that do not, in my

opinion, require any further round of review.

I list these minor remarks below, in order of appearance in the article, and do my best to help the authors dealing (or not dealing) with them as fast and easily as possible.

Introduction:

Lines 74-77: those two sentences imply that a shallower NADW & expanded AABW seen in data are not compatible with an increase in AMOC seen in models. My understanding is that both are not necessarily incompatible. It is e.g. discussed in Sigman et al., 2010, Nature (doi:10.1038/nature09149) in the chapter 'polar ventilation of the deep ocean', please check and verify that using the formulation 'This is at odds' is appropriate here.

3.3. Hydrological cycle:

Figure 6 indicates a large zonal changes btw PMIP4 & PMIP3 in the equatorial zone with increased (decreased) precip in PMIP4 wrt PMIP3 around Indonesia and Atlantic sectors (equatorial Pacific and western Indian ocean). It is complex, I agree, but I find this pattern very interesting, could you discuss a little bit more such pattern?

Sentence lines 282-284: again this feature is very interesting. Do you have any idea about what could cause that at first order? (shifts in ITCZ patterns, Sunda-Sahul shelf shelf exposure, ice sheet geometry?)

4 Data-model comparison:

I think you should immediately start by emphasizing that the new data reconstruction overall agree better with model outputs than the Bartlein and MARGO ones.

Lines 294-295: it seems that summer temperatures at high latitudes as estimated in the old and new datasets have nothing in common, please point it out.

Lines 318-319 (whereas the currently available PMIP4-CMIP6 simulations tend to be warmer than the reconstructions.): I don't really see that

Line 323: iLOVECLIM, did you mean AWIESM?

Lines 328-329 (However, the simulated change in winter temperature is smaller than indicated by the reconstructions (Fig. 12, top line).): not everywhere when the new datasets are considered

Lines 365-373 & Figure 13: Why such a figure? To me, it only shows improvement of Tierney in how the SST databasemight be representative if models get it right. I am not sure this last figure and paragraph is really helpful here.

Conclusion (The MARGO (2009) data set does not provide a strong constraint on the upper limit of the cooling because no model simulates warmer temperature anomalies than these reconstructions. ): even if I completely agree with you, you could not say that the MARGO data are wrong because models can't reproduce it. Please reformulate.

'Volodin 2018' in Table 1 is missing in the references

Figure 3: correct the caption 'same as Figure 1' (not 2). Also, is there something going on in the Walker circulation?

Figure 7, top left: please add the zero horizontal line, this will help

Figure 9: please add the 1:1 line and indicate which axis corresponds to which dataset

Figure 12: I wonder if is it technically manageable (and useful to the reader) to make a clear figure with the same Y axis for the same regions (and/or seasons), so we could appreciate the magnitude of the differences btw them?

---

## Short Comment (SC1) · 30 May 2020

The following metrics (multi-model mean) will be very useful for comparison with proxy and simulated large-scale temperatures from other PMIP/deepMIP paleo reference periods:

For each of six 30-deg latitude bands (18 values total, each with 5-95% uncertainties):

- ∆MAT

- ∆MAT over land

- ∆MAT over sea

This will enable polar amplification to calculated as 60-90 vs 90-90 latitude in both hemispheres, and land vs sea for both 90-90 and 60-60 or 30-30 lat, while providing a framework for comparisons with proxy data (e.g., Fig 4b in Brierley et al. https://doi.org/10.5194/cp-2019-168). For anomalies, please use 1850-1900 as the reference period.

Thank you in advance for including these key metrics in your paper.

---

## Referee Comment (RC2) · Anonymous Referee #2 · 10 Jun 2020

The manuscript provides a first evaluation of the Last Glacial Maximum (LGM) simulations performed for PMIP4-CMIP6 and a comparison with PMIP3-CMIP5. The authors focus on evaluating changes in temperature, the hydrological cycle, as well as atmospheric and oceanic circulation between PI and LGM simulations. The manuscript also provides an overview over how simulations results changed between these two rounds of PMIP. For their model-data comparisons they use new proxy data compilations as compared to the previous round of PMIP. This manuscript will provide a useful reference for a wide audience in the field. I did not find any major shortcomings with the manuscript, and I recommend this manuscript for publication after a few minor and technical comments and suggestions as detailed below have been addressed by the

authors.

Main points

1. The model ensembles in PMIP3 and PMIP4 are very different, only two modelling groups participating in PMIP3 had submitted LGM simulations from their new model versions to PMIP4. This is not the fault of the authors, but certain conclusions cannot be drawn in this situation. E.g. I don't think that the statement (abstract, lines 6-8) that "PMIP4-CMIP6 are globally less cold and less dry than the PMIP3-CMIP5 simulations, most probably because of the use of a more realistic specification of the northern hemisphere ice sheets in the latest simulations" is justified. This could be tested by simulating the new and old ice sheet configuration using the same models, but certainly not by just averaging two very different model ensembles. Similarly, in lines 281-282 "This, together with colder temperatures, could help explain why the PMIP4-CMIP6 models simulate a stronger AMOC at the LGM." Here at least the wording is more appropriate ("could help explain" instead of "most probably"), but I still think that this could very well be due to completely different reasons (AMOC is sensitive to a lot of things in different models). In summary, I believe the comparison PMIP3-PMIP4 is of limited use when it comes to attributing causes and/or processes to different outcomes, and I think it would strengthen the manuscript if the authors would make this more explicit, e.g. by mentioning this in the introduction and/or the methods section, and revise the statements mentioned above (and further lines 267-269; lines 342-344; lines 382-384; lines 388-389).

2. Given that the PMIP ensembles are relatively small anyway, it is a bit disturbing that not all PMIP4 models seem to have delivered all diagnostics, particularly since AMOC and northward heat transport are pretty standard, and all modelling groups are represented as co-authors on this manuscript as far as I can see. I would urge the authors to get the missing data in place for a revised version of a manuscript. Otherwise, in lines 227 and 240-241, I would not list the models that provide the data, but those who are missing (because they are fewer) and why.

3. Outliers: The authors are classifying some model results as outliers, apparently without any objective criterion. Why is CNRM-CM5 an outlier regarding tropical temperature change (line 198, Fig. 2)? The model is consistently warm over all latitude bands, and there are PMIP4 models that show almost the same weak cooling in the tropics. Likewise (lines 270-271, Fig. 7), why is CNRM-CM5 an outlier? I see that iLOVECLIM has a very strong reduction of precipitation in the tropics (is this due to the reduced complexity of the model compared to the others?). On the other hand, the zonal mean behaviour of INM-CM4 looks much stranger, so why isn't this model an outlier?

4. The description of changes in AMOC (lines 228-233) is a bit unclear: "Two of the PMIP4-CMIP6 models show a deep NADW cell reaching the ocean floor in the North Atlantic, whereas four of the PMIP4-CMIP6 models simulate some Antarctic Bottom Water (AABW) in the North Atlantic." The AWIESM1 and the MIROC-ES2L have a very deep cell already in the PI. The only model showing a strong change is iLOVECLIM. What do the authors mean by "some AABW". Likewise: "The intrusion of AABW cell into the North Atlantic was shown by some of the PMIP3-CMIP5 simulations, but not as much as the PMIP4-CMIP6 simulations". I have difficulties seeing this in Fig.4. In PMIP3, only CCSM4 shows an "intrusion of AABW" from PI to LGM, and in PMIP4 I do not see this at all(?). If there is a clear change of AABW intrusion between PMIP3/4 maybe an additional figure could help? Please consider revising the description of overturning changes.

Other points:

line 47-48: "...several of the PMIP4-CMIP6 models having substantially higher climate sensitivity than the PMIP3-CMIP5 versions of the same models, and thus the range of climate sensitivity sampled by the PMIP4-CMIP6 models is much wider." It should be mentioned already here, that this is not true for the models actually included in this study (as stated further below, lines 122-125).

[Figure]

lines 159-160: I do not understand "...we only use the data synthesis for comparisons here." Is other data used for other purposes in this work? Please clarify.

lines 187-188: "As expected, the simulations show larger changes over the land than over ocean." It would be interesting to read how much (i.e. maybe state the average land-sea gradient for PI and LGM?)

lines 358-359: "The PMIP4-CMIP6 models are more consistent with the temperature reconstructions over tropical Asia, but show poorer agreement with the precipitation reconstructions than the PMIP3-CMIP5 models." I would say "slightly more consistent". As to the precipitation reconstructions over tropical Asia, it could be mentioned that PMIP4 models at least agree on the sign of change in contrast to PMIP3.

Please complete Table 1 as far as possible (some references are missing, and it should be possible to collect information on PMIP3 spinup duration at least for MPI and MIROC).

Technical:

lines 12-13: "...remain large so,..." (?) -> maybe "...remain large, and although..."

line 28: delete "rise"

lines 34-35: "Atmospheric greenhouse gases (GHGs) were lower than pre-industrial (PI) values,..." -> "Atmospheric greenhouse gas (GHG) concentrations were lower than during the pre-industrial (PI) period,..."

line 83: alternate -> alternative (?)

line 206: the authors probably mean "significant" not "real".

line 332: "Although" does not make sense, consider rewording the beginning of this sentence.

---

## Author Comment (AC2) · 29 Oct 2020

**Response to reviewers**

For easier reading, we repeat the reviewer's comments in black and answer then in *blue*.

**Reviewer 2**

The manuscript provides a first evaluation of the Last Glacial Maximum (LGM) simulations performed for PMIP4-CMIP6 and a comparison with PMIP3-CMIP5. The authors focus on evaluating changes in temperature, the hydrological cycle, as well as atmospheric and oceanic circulation between PI and LGM simulations. The manuscript also provides an overview over how simulations results changed between these two rounds of PMIP. For their model-data comparisons they use new proxy data compilations as compared to the previous round of PMIP. This manuscript will provide a useful reference for a wide audience in the field. I did not find any major shortcomings with the manuscript, and I recommend this manuscript for publication after a few minor and technical comments and suggestions as detailed below have been addressed by the authors.
*We thank the reviewer for these supportive comments.*

Main points

1. The model ensembles in PMIP3 and PMIP4 are very different, only two modelling groups participating in PMIP3 had submitted LGM simulations from their new model versions to PMIP4. This is not the fault of the authors, but certain conclusions cannot be drawn in this situation. E.g. I don't think that the statement (abstract, lines 6-8) that "PMIP4-CMIP6 are globally less cold and less dry than the PMIP3-CMIP5 simulations, most probably because of the use of a more realistic specification of the northern hemisphere ice sheets in the latest simulations" is justified. This could be tested by simulating the new and old ice sheet configuration using the same models, but certainly not by just averaging two very different model ensembles. Similarly, in lines 281-282 "This, together with colder temperatures, could help explain why the PMIP4-CMIP6 models simulate a stronger AMOC at the LGM." Here at least the wording is more appropriate ("could help explain" instead of "most probably"), but I still think that this could very well be due to completely different reasons (AMOC is sensitive to a lot of things in different models). In summary, I believe the comparison PMIP3-PMIP4 is of limited use when it comes to attributing causes and/or processes to different outcomes, and I think it would strengthen the manuscript if the authors would make this more explicit, e.g. by mentioning this in the introduction and/or the methods section, and revise the statements mentioned above (and further lines 267-269; lines 342-344; lines 382-384; lines 388-389).
*The results of two new models, IPSLCM5A2 and CESM1.2, for which previous versions existed in PMIP3. However, we agree that the models in the PMIP3 and PMIP4 sets are not simply different versions of the same models. We will modify the text to highlight this. This is also shown by Figure 1, which simply displays the difference in mean annual precipitation for PI, between the PMIP3 and PMIP4 ensembles.*

2. Given that the PMIP ensembles are relatively small anyway, it is a bit disturbing that not all PMIP4 models seem to have delivered all diagnostics, particularly since AMOC and northward heat transport are pretty standard, and all modelling groups are represented as co-authors on this manuscript as far as I can see. I would urge the authors to get the missing data in place for a revised version of a manuscript. Otherwise, in lines 227 and 240-241, I would not list the models that provide the data, but those who are missing (because they are fewer) and why.
*We will attempt to add the missing diagnostics in for the next version of the manuscript and will modify the text as suggested. We actually have collected a lot of the missing data.*

3. Outliers: The authors are classifying some model results as outliers, apparently without any objective criterion. Why is CNRM-CM5 an outlier regarding tropical temperature change (line 198,

Fig. 2)? The model is consistently warm over all latitude bands, and there are PMIP4 models that show almost the same weak cooling in the tropics. Likewise (lines 270-271, Fig. 7), why is CNRM-CM5 an outlier? I see that iLOVECLIM has a very strong reduction of precipitation in the tropics (is this due to the reduced complexity of the model compared to the others?). On the other hand, the zonal mean behaviour of INM-CM4 looks much stranger, so why isn't this model an outlier?

*We agree that we used the word "outlier" too loosely here. CNRM appears to stand out from the PMIP3 model results for the LGM tropical temperatures (Fig 2) but this is not seen on the zonal mean plot because many PMIP4 models mostly stand between CNRM and the other PMIP3 models. We will attempt to use the word more precisely in the next version of the manuscript, in particular stating which ensemble we consider when pointing to an "outlier".*

4. The description of changes in AMOC (lines 228-233) is a bit unclear: "Two of the PMIP4-CMIP6 models show a deep NADW cell reaching the ocean floor in the North Atlantic, whereas four of the PMIP4-CMIP6 models simulate some Antarctic Bottom Water (AABW) in the North Atlantic." The AWIESM1 and the MIROC-ES2L have a very deep cell already in the PI. The only model showing a strong change is iLOVECLIM. What do the authors mean by "some AABW". Likewise: "The intrusion of AABW cell into the North Atlantic was shown by some of the PMIP3-CMIP5 simulations, but not as much as the PMIP4-CMIP6 simulations". I have difficulties seeing this in Fig.4. In PMIP3, only CCSM4 shows an "intrusion of AABW" from PI to LGM, and in PMIP4 I do not see this at all(?). If there is a clear change of AABW intrusion between PMIP3/4 maybe an additional figure could help? Please consider revising the description of overturning changes.

*We agree that this description is not very precise and will update it in the next version of the manuscript. In particular we will add the models' names after the assertions and add the definitions of NADW and AABW on the plots. We will also include the notions of change in ocean circulation state, like for iLOVECLIM, to descriptions of PI and LGM states. We also plan to add an analysis of AMOC strength, AABW strength, and AMOC depth.*

Other points:

line 47-48: "...several of the PMIP4-CMIP6 models having substantially higher climate sensitivity than the PMIP3-CMIP5 versions of the same models, and thus the range of climate sensitivity sampled by the PMIP4-CMIP6 models is much wider." It should be mentioned already here, that this is not true for the models actually included in this study (as stated further below, lines 122-125).

*This is right and will be added in the text. We were actually hoping to include those models at a second stage, but this probably will not be the case.*

lines 159-160: I do not understand "...we only use the data synthesis for comparisons here." Is other data used for other purposes in this work? Please clarify.

*We meant that we only use the information at data sites, not the global result from the data assimilation. In the new version of the paper, we will actually show both, which raises interesting points about representativity of the sites.*

lines 187-188: "As expected, the simulations show larger changes over the land than over ocean." It would be interesting to read how much (i.e. maybe state the average land-sea gradient for PI and LGM?).

*Fine, we will add the quantification in the next version of the manuscript, by computing the regression between the two quantities.*

lines 358-359: "The PMIP4-CMIP6 models are more consistent with the temperature reconstructions over tropical Asia, but show poorer agreement with the precipitation reconstructions than the PMIP3-CMIP5 models." I would say "slightly more consistent". As to the precipitation reconstructions

over tropical Asia, it could be mentioned that PMIP4 models at least agree on the sign of change in contrast to PMIP3.
*It turns out that this conclusion was drawn from a very few points over this region, so we have removed it from the new version of Figure 12.*

Please complete Table 1 as far as possible (some references are missing, and it should be possible to collect information on PMIP3 spinup duration at least for MPI and MIROC).
*Table 1 is now much more complete, in particular for the PMIP4 models which are documented here altogether for the first time.*

Technical:

lines 12-13: "...remain large so,..." (?) -> maybe "...remain large, and although..."
*This has been corrected, thank you.*

line 28: delete "rise"
*OK, done.*

lines 34-35: "Atmospheric greenhouse gases (GHGs) were lower than pre-industrial (PI) values,..." -> "Atmospheric greenhouse gas (GHG) concentrations were lower than during the pre-industrial (PI) period,..."
*OK, done.*

line 83: alternate -> alternative (?)
*yes, this is what we meant.*

line 206: the authors probably mean "significant" not "real".
*We will reformulate the sentence.*

line 332: "Although" does not make sense, consider rewording the beginning of this sentence.
*This is right. The sentence will be modified, taking into account an updated method to quantify model-related variability.*

---

## Author Comment (AC3) · 29 Oct 2020

These measures will be provided along with the manuscript.

---

## Author Response (AR1)

**Letter to the Editor**

Dear Editor,

Here is, at last, the revised version of the manuscript. We have updated the manuscript with the latest simulations as well as with all the data syntheses we knew of that are more recent than those used for the PMIP3-CMIP5 simulations (i.e. Bartlein et al. (2011) and MARGO (2009). This results in a more complete manuscript with revisions going beyond those asked by the reviewers, but we aimed at providing a manuscript as up-to-date as possible. We have therefore also updated our responses to the reviewers below.

Best regards

Masa Kageyama, on behalf of the authors (which are also more numerous than for the first submission)

**Response to reviewers**

For easier reading, we repeat the reviewer's comments in black and answer then in *blue*.

**Reviewer 1**

First of all, I apologise to the authors for my late review.
*We apologise for the late revision of the manuscript! The new version of the manuscript includes updated reconstructions as well as the results from 3 new models (5 simulations). This results in modifications of some of the manuscript's main messages but we felt it was important to keep the results as up to date as possible.*

The article presents preliminary results of the new PMIP4-CMIP6 exercise, compare it to the former PMIP3-CMIP5 simulation ensemble, along with new LGM temperature databases for continental and oceanic temperatures. It is an important and interesting update of the LGM experiments, well organized, and provides a state-of the art on recent advances in climate modeling and data compilation. It is very descriptive but I enjoyed very much reading it. Sometimes, figures are not easy to read, but I don't see how it could be improved given all the dimensions the model/data comparison requests. I recommend publication with some minor rectifications that do not, in my opinion, require any further round of review.
*We thank the reviewer for his positive comments.*

I list these minor remarks below, in order of appearance in the article, and do my best to help the authors dealing (or not dealing) with them as fast and easily as possible.

Introduction:

Lines 74-77: those two sentences imply that a shallower NADW & expanded AABW seen in data are not compatible with an increase in AMOC seen in models. My understanding is that both are not necessarily incompatible. It is e.g. discussed in Sigman et al., 2010, Nature (doi:10.1038/nature09149) in the chapter 'polar ventilation of the deep ocean', please check and verify that using the formulation 'This is at odds' is appropriate here.
*We actually meant "deepening of the AMOC", and have corrected the manuscript accordingly. We have also completed the list of references. The corresponding paragraph now reads:*
*"The LGM boundary conditions also had a strong impact on ocean circulation, as documented via multiple tracers (e.g. Lynch-Stieglitz et al., 2007, Jaccard and Galbraith, 2011, Böhm et al., 2015) which suggest a shallower North Atlantic Deep Water cell and expanded Antarctic Bottom Water*

*(AABW). Besides, Gebbie (2014) uses a combination of synthesis of multiple tracers measured in sediment cores for the LGM and a global tracer transport model to show that these tracers are compatible with a vertical distribution of NADW and AABW similar to today, but that the core of the NADW water mass shoals by 1000m. None of these proposed reconstructions of glacial circulation is consistent with the PMIP3-CMIP5 model results (Muglia and Schmittner, 2015) which all show a deepening of the Atlantic Meridional Overturning Circulation (AMOC), with NADW reaching the ocean floor in the northern North Atlantic for some models."*

3.3. Hydrological cycle:

Figure 6 indicates a large zonal changes btw PMIP4 & PMIP3 in the equatorial zone with increased (decreased) precip in PMIP4 wrt PMIP3 around Indonesia and Atlantic sectors (equatorial Pacific and western Indian ocean). It is complex, I agree, but I find this pattern very interesting, could you discuss a little bit more such pattern?
Sentence lines 282-284: again this feature is very interesting. Do you have any idea about what could cause that at first order? (shifts in ITCZ patterns, Sunda-Sahul shelf shelf exposure, ice sheet geometry?)

*Yes, these patterns are standing out from the maps showing PMIP4 – PMIP3 differences in precipitation and in precipitation minus evaporation. The pattern is actually similar for PI (cf Fig. 1 below), suggesting that it is the PMIP4 models which perform differently from the PMIP3 ones, and not so much a difference in the responses to LGM forcings and boundary conditions. We have added this figure as a panel and Fig 6 and have included this new item in our discussion of the changes in the hydrological cycle.*

*Figure 1 : Differences in mean annual precipitation simulated by the PMIP4 models (ensemble mean), compared to the PMIP3 models (ensemble mean).*

4 Data-model comparison:

I think you should immediately start by emphasizing that the new data reconstruction overall agree better with model outputs than the Bartlein and MARGO ones.
*With updated reconstructions compared to the first submission of this manuscript, this is not quite true anymore. We have therefore fully updated the first paragraphs of this section.*

Lines 294-295: it seems that summer temperatures at high latitudes as estimated in the old and new datasets have nothing in common, please point it out.

*We disagree that summer temperatures at high latitudes are substantially different in the two data sets. There is more scatter in the MTWA reconstructions from Bartlein and this is what causes the apparent shift from the 1:1 line at the cool end. However, there are certainly points where there is congruence between the two sets of reconstructions. Furthermore, there is no significant difference in GDD, an alternative measure of summer warmth, between the two data sets. We have made a systematic comparison of the datasets by latitude, shown below (Figure 2), which indicates that there is overlap in the reconstructions, though as we state Cleator tends to reconstruct less extreme cold temperatures and be less noisy - which is a consequence of the reconstruction technique. A more complete figure in the style of Figure 2 below is now included in the manuscript, replacing Fig 9, to better compare the different continental reconstructions.*

[Figure]

*Figure 2 : comparison of the reconstructed anomalies for MTCO, MTWA and MAP, averaged by latitudinal bands. Bartlein et al. (2011) reconstructions in orange, Cleator et al. (2019) reconstructions, only at points where there is a value for the Bartlein et al. (2011) reconstruction In dark blue, full Cleator et al. (2019) reconstruction in green.*

Lines 318-319 (whereas the currently available PMIP4-CMIP6 simulations tend to be warmer than the reconstructions.): I don't really see that

*Right. This sentence should have been "whereas the currently available PMIP4-PMIP6 simulations tend to stand closer to the reconstructions or on their warm side. " However, with the inclusion of the results from two new models (CESM1.2 and IPSLCM5A2) this sentence might have to modified.*

Line 323: iLOVECLIM, did you mean AWIESM?

*Indeed, we meant AWIESM2. But we removed this complex sentence after the inclusion of more models and more reconstructions.*

Lines 328-329 (However, the simulated change in winter temperature is smaller than indicated by the reconstructions (Fig. 12, top line).): not everywhere when the new datasets are considered
*This is actually strictly right only for western Europe. This paragraph has been updated quite a lot given the results of the new models, but the discrepancy for western Europe winter temperature remains and is now stated more clearly..*

Lines 365-373 & Figure 13: Why such a figure? To me, it only shows improvement of Tierney in how the SST database might be representative if models get it right. I am not sure this last figure and paragraph is really helpful here.
*This figure is an attempt to constrain the global mean surface temperature change based on the different data sets. We have tried to make this clearer in the new version of the manuscript.*

Conclusion (The MARGO (2009) data set does not provide a strong constraint on the upper limit of the cooling because no model simulates warmer temperature anomalies than these reconstructions.): even if I completely agree with you, you could not say that the MARGO data are wrong because models can't reproduce it. Please reformulate.
*We have reformulated this sentence as: " Using the terrestrial data sets as a constraint, indicates a global cooling between -6.7 and -3.7°C, while using Tierney et al. (2020) as a constraint indicates a global cooling of -4.9 to -3.2°C and using the MARGO (2009) and GLOMAP2020 datasets constrain the global average to be above -3.9°C and -4.4°C, respectively."*

'Volodin 2018' in Table 1 is missing in the references.
*This reference has been added in the revised version of the manuscript.*

Figure 3: correct the caption 'same as Figure 1' (not 2).
*This has been corrected, thank you.*
Also, is there something going on in the Walker circulation?
*It is difficult to draw strong conclusions on the Walker circulation from this figure only. This topic is left for further study.*

Figure 7, top left: please add the zero horizontal line, this will help
*OK, this is now done for figures 2 and 7, where we have also added the ensemble mean anomaly on the regional average plots.*

Figure 9: please add the 1:1 line and indicate which axis corresponds to which dataset
*We have actually changed the design of this figure, in order to compare the data sets for different bands of latitude.*

Figure 12: I wonder if is it technically manageable (and useful to the reader) to make a clear figure with the same Y axis for the same regions (and/or seasons), so we could appreciate the magnitude of the differences btw them?
*We have updated the figure so that the scales are more uniform, and have added the PMIP3 and PMIP4 means for easier discussion. We have also removed the panels for Tropical Asia,which were only based on a few data sites, and added the comparison to the alternative datasets for the tropical ocean temperatures to completeness.*

**Reviewer 2**

The manuscript provides a first evaluation of the Last Glacial Maximum (LGM) simulations performed for PMIP4-CMIP6 and a comparison with PMIP3-CMIP5. The authors focus on evaluating changes in temperature, the hydrological cycle, as well as atmospheric and oceanic circulation between PI and LGM simulations. The manuscript also provides an overview over how simulations results changed between these two rounds of PMIP. For their model-data comparisons they use new proxy data compilations as compared to the previous round of PMIP. This manuscript will provide a useful reference for a wide audience in the field. I did not find any major shortcomings with the manuscript, and I recommend this manuscript for publication after a few minor and technical comments and suggestions as detailed below have been addressed by the authors.
*We thank the reviewer for these supportive comments.*

Main points

1. The model ensembles in PMIP3 and PMIP4 are very different, only two modelling groups participating in PMIP3 had submitted LGM simulations from their new model versions to PMIP4. This is not the fault of the authors, but certain conclusions cannot be drawn in this situation. E.g. I don't think that the statement (abstract, lines 6-8) that "PMIP4-CMIP6 are globally less cold and less dry than the PMIP3-CMIP5 simulations, most probably because of the use of a more realistic specification of the northern hemisphere ice sheets in the latest simulations" is justified. This could be tested by simulating the new and old ice sheet configuration using the same models, but certainly not by just averaging two very different model ensembles. Similarly, in lines 281-282 "This, together with colder temperatures, could help explain why the PMIP4-CMIP6 models simulate a stronger AMOC at the LGM." Here at least the wording is more appropriate ("could help explain" instead of "most probably"), but I still think that this could very well be due to completely different reasons (AMOC is sensitive to a lot of things in different models). In summary, I believe the comparison PMIP3-PMIP4 is of limited use when it comes to attributing causes and/or processes to different outcomes, and I think it would strengthen the manuscript if the authors would make this more explicit, e.g. by mentioning this in the introduction and/or the methods section, and revise the statements mentioned above (and further lines 267-269; lines 342-344; lines 382-384; lines 388-389).
*The results of two new models, IPSLCM5A2 and CESM1.2, for which previous versions existed in PMIP3. However, we agree that the models in the PMIP3 and PMIP4 sets are not simply different versions of the same models. We modified the text to highlight this, at the end of section 2.2 and at the end of section 3.1 on the results in terms of mean annual temperature. For the hydrological cycle (section 3.3) we have included a new panel in figure 6 (also shown in the response to Reviewer 1), which simply displays the difference in the simulated mean annual precipitation for PI, between the PMIP3 and PMIP4 ensembles.*

2. Given that the PMIP ensembles are relatively small anyway, it is a bit disturbing that not all PMIP4 models seem to have delivered all diagnostics, particularly since AMOC and northward heat transport are pretty standard, and all modelling groups are represented as co-authors on this manuscript as far as I can see. I would urge the authors to get the missing data in place for a revised version of a manuscript. Otherwise, in lines 227 and 240-241, I would not list the models that provide the data, but those who are missing (because they are fewer) and why.
*We have now gathered the results from more models, although some are still missing for this section.*

3. Outliers: The authors are classifying some model results as outliers, apparently without any objective criterion. Why is CNRM-CM5 an outlier regarding tropical temperature change (line 198, Fig. 2)? The model is consistently warm over all latitude bands, and there are PMIP4 models that show almost the same weak cooling in the tropics. Likewise (lines 270-271, Fig. 7), why is CNRM-CM5 an outlier? I see that iLOVECLIM has a very strong reduction of precipitation in the tropics (is this due

to the reduced complexity of the model compared to the others?). On the other hand, the zonal mean behaviour of INM-CM4 looks much stranger, so why isn't this model an outlier?

*We agree that we used the word "outlier" too loosely here. With the results of more models, we have updated many paragraphs and avoided the use of this word, which is not appropriate give the new results.*

4. The description of changes in AMOC (lines 228-233) is a bit unclear: "Two of the PMIP4-CMIP6 models show a deep NADW cell reaching the ocean floor in the North Atlantic, whereas four of the PMIP4-CMIP6 models simulate some Antarctic Bottom Water (AABW) in the North Atlantic." The AWIESM1 and the MIROC-ES2L have a very deep cell already in the PI. The only model showing a strong change is iLOVECLIM. What do the authors mean by "some AABW". Likewise: "The intrusion of AABW cell into the North Atlantic was shown by some of the PMIP3-CMIP5 simulations, but not as much as the PMIP4-CMIP6 simulations". I have difficulties seeing this in Fig.4. In PMIP3, only CCSM4 shows an "intrusion of AABW" from PI to LGM, and in PMIP4 I do not see this at all(?). If there is a clear change of AABW intrusion between PMIP3/4 maybe an additional figure could help? Please consider revising the description of overturning changes.

*We agree that this description was not precise and have updated it in the new version of the manuscript. In particular we have added the definitions of NADW and AABW (both in the text and in Fig. 4's caption). We have also added the models' names after the assertions and described the changes in NADW depth and strength more systematically.*

Other points:

line 47-48: "...several of the PMIP4-CMIP6 models having substantially higher climate sensitivity than the PMIP3-CMIP5 versions of the same models, and thus the range of climate sensitivity sampled by the PMIP4-CMIP6 models is much wider." It should be mentioned already here, that this is not true for the models actually included in this study (as stated further below, lines 122-125).

*This is right and has been added in the text. We were actually hoping to include those models at a second stage, but this is not the case.*

lines 159-160: I do not understand "...we only use the data synthesis for comparisons here." Is other data used for other purposes in this work? Please clarify.

*We meant that we only use the information at data sites, not the global result from the data assimilation. In the new version of the paper, we will actually show both, which raises interesting points about representativity of the sites.*

lines 187-188: "As expected, the simulations show larger changes over the land than over ocean." It would be interesting to read how much (i.e. maybe state the average land-sea gradient for PI and LGM?).

*Fine, we have added the quantification, for the global temperatures as well as for the tropics.*

lines 358-359: "The PMIP4-CMIP6 models are more consistent with the temperature reconstructions over tropical Asia, but show poorer agreement with the precipitation reconstructions than the PMIP3-CMIP5 models." I would say "slightly more consistent". As to the precipitation reconstructions over tropical Asia, it could be mentioned that PMIP4 models at least agree on the sign of change in contrast to PMIP3.

*It turns out that this conclusion was drawn from a very few points over this region, so we have removed it from the new version of Figure 12.*

Please complete Table 1 as far as possible (some references are missing, and it should be possible to collect information on PMIP3 spinup duration at least for MPI and MIROC).

*Table 1 is now much more complete, in particular for the PMIP4 models which are documented here altogether for the first time.*

Technical:

lines 12-13: "...remain large so,..." (?) -> maybe "...remain large, and although..."
*This has been corrected, thank you.*

line 28: delete "rise"
*OK, done.*

lines 34-35: "Atmospheric greenhouse gases (GHGs) were lower than pre-industrial (PI) values,..." -> "Atmospheric greenhouse gas (GHG) concentrations were lower than during the pre-industrial (PI) period,..."
*OK, done.*

line 83: alternate -> alternative (?)
*yes, this is what we meant.*

line 206: the authors probably mean "significant" not "real".
*This sentence has been removed.*

line 332: "Although" does not make sense, consider rewording the beginning of this sentence.
*This is right. The sentence will be modified, taking into our new results.*

**Short Comment 1**

The following metrics (multi-model mean) will be very useful for comparison with proxy and simulated large-scale temperatures from other PMIP/deepMIP paleo reference periods:
For each of six 30-deg latitude bands (18 values total, each with 5-95% uncertainties):
- _MAT
- _MAT over land
- _MAT over sea

This will enable polar amplification to calculated as 60-90 vs 90-90 latitude in both hemispheres, and land vs sea for both 90-90 and 60-60 or 30-30 lat, while providing a framework for comparisons with proxy data (e.g., Fig 4b in Brierley et al. https://doi.org/10.5194/cp-2019-168). For anomalies, please use 1850-1900 as the reference period.
Thank you in advance for including these key metrics in your paper.

*We have included the averages over 30° latitude bands for the mean annual surface air temperature over the land, over the oceans, and over both land and oceans, for each model, in the supplementary material (Tables S2 to S4).*